# Bioengineered Efficacy Models of Skin Disease: Advances in the Last 10 Years

**DOI:** 10.3390/pharmaceutics14020319

**Published:** 2022-01-28

**Authors:** Diana Nicole Stanton, Gitali Ganguli-Indra, Arup Kumar Indra, Pankaj Karande

**Affiliations:** 1Department of Biomedical Engineering, Rensselaer Polytechnic Institute, Troy, NY 12180, USA; stantd3@rpi.edu; 2Department of Chemical and Biological Engineering, Rensselaer Polytechnic Institute, Troy, NY 12180, USA; 3Department of Pharmaceutical Sciences, College of Pharmacy, Oregon State University, Corvallis, OR 97331, USA; Gitali.Indra@oregonstate.edu (G.G.-I.); Arup.Indra@oregonstate.edu (A.K.I.); 4Knight Cancer Institute, Oregon Health and Science University, Portland, OR 97239, USA; 5Department of Biochemistry and Biophysics, Oregon State University, Corvallis, OR 97331, USA; 6Linus Pauling Science Center, Oregon State University, Corvallis, OR 97331, USA; 7Department of Dermatology, Oregon Health and Science University, Portland, OR 97239, USA

**Keywords:** skin, skin disease, human skin equivalents, 3D bioprinting, disease modeling, skin disease modeling

## Abstract

Models of skin diseases, such as psoriasis and scleroderma, must accurately recapitulate the complex microenvironment of human skin to provide an efficacious platform for investigation of skin diseases. Skin disease research has been shifting from less complex and less relevant 2D (two-dimensional) models to significantly more relevant 3D (three-dimensional) models. Three-dimensional modeling systems are better able to recapitulate the complex cell–cell and cell–matrix interactions that occur in vivo within skin. Three-dimensional human skin equivalents (HSEs) have emerged as an advantageous tool for the study of skin disease in vitro. These 3D HSEs can be highly complex, containing both epidermal and dermal compartments with integrated adnexal structures. The addition of adnexal structures to 3D HSEs has allowed researchers to gain more insight into the complex pathology of various hereditary and acquired skin diseases. One method of constructing 3D HSEs, 3D bioprinting, has emerged as a versatile and useful tool for generating highly complex HSEs. The development of commercially available 3D bioprinters has allowed researchers to create highly reproducible 3D HSEs with precise integration of multiple adnexal structures. While the field of bioengineered models for study of skin disease has made tremendous progress in the last decade, there are still significant efforts necessary to create truly biomimetic skin disease models. In future studies utilizing 3D HSEs, emphasis must be placed on integrating all adnexal structures relevant to the skin disease under investigation. Thorough investigation of the intricate pathology of skin diseases and the development of effective treatments requires use of highly efficacious models of skin diseases.

## 1. Introduction

Globally, skin diseases contribute 1.79% of the overall burden of disease as measured using disability-adjusted life years (DALYs) [1]. Atopic dermatitis (AD), psoriasis, and melanoma are among the skin diseases with the highest global DALYs [1]. In the United States alone, one in four Americans were seen by a physician for one or more skin diseases in 2013 [2]. Skin diseases cost the United States healthcare system $75 billion in direct costs and $11 billion in indirect lost opportunity costs in 2013 [3]. The mean capitalized research and development investment required to bring a new drug to market is estimated at $1335.9 million and an average of 7 to 12 years from preclinical testing to drug approval [4,5]. In addition to this, only 19% of drugs that enter clinical trials ultimately receive approval from the FDA [5]. Given the considerable financial and temporal investments required to develop a treatment, it is critical that the most accurate in vitro models be used during the initial stages of preclinical research used to develop new treatments.

While 2D models have provided the basis for much of our modern understanding of biology, they fail to accurately recapitulate the complex in vivo 3D microenvironment [6]. Three-dimensional modeling has shown significant potential to recreate cell fates observed in vivo. Whether a 2D or 3D model is used in a study can significantly impact cell proliferation, cell differentiation, mechanical responses of cells, and cell survival [7,8,9]. Use of animal models is required for preclinical trials by the Nuremburg Code; however, there are questions as to how accurate these models are for modeling human processes [10,11]. In transgenic animal models designed to mimic human hereditary diseases, there have been numerous instances of significant differences between the display of phenotypes observed in animal models and humans [11].

Because of the shortcomings of 2D in vitro models and animal models, 3D models in preclinical trials have become an increasingly attractive alternative. In skin disease modeling, these 3D models can take the form of 3D HSEs. These models can provide many of the biochemical and biomechanical cues found in native human skin that play a key role in skin disease pathogenesis. Three-dimensional HSEs are highly tunable and can vary significantly in their level of sophistication depending on the desired application of the model. While these models do provide notable advantages over 2D in vitro models and animal models, there are still significant improvements to these models that must be made. Increasingly sophisticated, highly biomimetic 3D HSEs will enable a more complete understanding of skin disease pathogenesis. This will enable researchers to develop more effective therapies for these skin diseases. Most skin disease models use a basic form of HSE generated using manual deposition of an epidermal compartment composed of keratinocytes and cell media over a dermal compartment composed of fibroblasts and collagen type I [12,13,14,15]. While these models do provide more insight into the pathogenesis of skin disease than a typical 2D in vitro model would, they fail to fully capture the intricacies of both healthy and diseased skin. To accurately recapitulate the pathogenesis of a disease, all relevant adnexa affected by the skin disease must be included. Investigation of the interplay between various skin diseases and skin adnexa will allow for a better understanding of the underlying pathology of different skin disease, which will in turn provide insights that can be applied to superior treatments of these diseases. 

This review focuses on the progress made towards accurately modeling skin diseases using 3D HSEs over the last decade and the development of different methods for producing these models. Particular emphasis is placed on the use of 3D bioprinting in construction of 3D HSEs. Existing 3D platforms for investigation of hereditary and acquired skin disease are highlighted, including those that exist for autoimmune skin diseases. The influence of genetic factor B-cell lymphoma/leukemia 11A (BCL11A) on skin barrier defects involved in AD and ichthyosis vulgaris is further expanded upon. This review examines what these existing 3D models have successfully captured, what aspects of the diseases discussed were not captured well, and how advances in the field on 3D skin modeling could improve these platforms and facilitate future development of effective treatments for skin diseases [16,17,18,19].

## 2. Overview of Skin Structure and Function

Skin is the largest organ of the body and performs many critical functions. It acts as a physical barrier between the body and the external environment [20]. It provides a first line of defense against invasion by microorganisms. It serves as a thermoregulatory system, controls water loss, and provides protection against ultraviolet light [20]. Skin also provides a means of receiving stimuli from the external environment, including the sensations of touch, temperature, itch (pruritus), pressure, and pain [21].

Skin is composed of the epidermis, dermis, hypodermis, and adnexa. The epidermis lies superficial to the dermis and is composed of four to five layers that are characterized by the morphological features of the keratinocytes in each layer [22]. Keratinocytes originate from germinal cells in the basal layer and undergo cornification to become corneocytes as they ascend to the stratum corneum [22]. The stratum corneum is the outermost layer of the epidermis and is the interface between the external and internal environment. It is composed of several sheets of flattened corneocytes embedded in a lipid matrix containing a dense network of keratin [22,23]. This layer acts as a barrier against invasion by microorganisms and regulates loss of water and salts through the skin. The keratinocytes of the stratum granulosum play a role in the barrier function and intercellular cohesion of the epidermis [24]. The keratinocytes of the stratum spinosum form a layer of polyhedral cells connected by desmosomes, allowing strong bonds to form between the keratinocytes. The deepest layer of the epidermis is the stratum basale and is responsible for the production of keratinocytes [22]. Melanocytes, pigment producing cells, are also present in the stratum basale.

The dermis is a layer of connective tissue that lies between the epidermis and hypodermis. The papillary dermis is the uppermost layer of the dermis and interdigitates with the rete ridges of the epidermis, serving to increase connection between the epidermal and dermal layers. Deep to the papillary dermis is the reticular dermis. The reticular dermis houses many of the skin’s adnexa and provides the skin with strength, elasticity, and flexibility [25]. The most abundant cells in the dermis are fibroblasts, which are responsible for the synthesis of collagen and elastin fibers found in the dermis [22]. The hypodermis lies deep to the dermis and connects the skin to the underlying fascia of the bones and muscles. 

The adnexa found in skin, shown in Figure 1, also play a crucial role in its ability to perform its functions. The vasculature found within skin travels from larger vessels through the hypodermis and forms loops in the papillary dermis. Capillaries extend from these loops to provide nutrients and remove waste from the epidermis [26]. The nerves found within the skin lie close to this vasculature and provide sensory input.

Hair follicles are found in the dermal and hypodermal layers of the skin and extend to the surface of the epidermis [27]. Hair follicles are complex enough to be considered mini organs and serve a variety of purposes, including thermal insulation, dispersion of sweat and sebum to the skin surface, and sensation. Hair follicles also contain a repository of stem cells within their bulbs, aiding in wound healing following insult to skin [27].

Glands found in the skin include sweat glands and sebaceous glands. Sweat glands fall into two primary categories, apocrine and eccrine glands. Eccrine sweat glands primarily secrete water and electrolytes onto the surface of skin. Apocrine glands are found in haired skin and secrete sebum, an oily substance containing lipids, proteins, and steroids. Sebum acts as a lubricant that protects skin against friction and aids in skin’s role as a permeability barrier [28]. Eccrine glands are responsible for temperature regulation and respond to changes in temperature. Apocrine glands respond to emotional stimuli [29].

## 3. Skin Diseases

Skin diseases occur when the intricate dynamics that allow skin to perform its many functions are disrupted. While there are many types of skin diseases, this review focuses on hereditary skin diseases, acquired skin diseases, and autoimmune skin diseases, which may be of either hereditary or acquired origin. Hereditary skin diseases are inheritable disorders that affect the skin. Hereditary skin diseases occur because of an abnormality in an individual’s genome [30]. This abnormality may be monogenic, affecting only one gene, or polygenic, affecting multiple genes [31]. Acquired skin diseases are disorders involving the skin that occur at some point throughout an individual’s life instead of being inherited and present at birth. Acquired skin diseases may occur because of a variety of causes, including environmental factors. Occupational skin diseases also fall into the category of acquired skin diseases, with the most common being contact dermatitis caused by exposure to skin irritants [32]. Autoimmune skin diseases are a group of inflammatory disorders characterized by the immunological abnormalities that play a crucial role in the diseases [33]. An autoimmune skin disease may be either hereditary, such as most forms of EB and ichthyosis, or acquired, such as epidermolysis bullosa acquisita (EBA) and atopic dermatitis (AD).

Epidermolysis bullosa (EB) is a hereditary skin disease characterized by structural fragility of the skin that causes recurrent blister formation. There are over 30 clinical subtypes of EB, with a minimum of 18 distinct genes affected [34,35]. EB is classified into four primary subtypes based primarily on the level of skin cleavage observed: EB simplex, junctional EB, dystrophic EB, and Kindler syndrome [35]. There is also a noninherited, autoimmune form of EB termed EBA [36]. EBA is induced by the presence of autoantibodies to type VII collagen, causing the formation of mucocutaneous blisters [37].

Ichthyosis is another example of a hereditary skin disease. The term ichthyosis is given to a large group of heterogenous cornification disorders [38]. Ichthyosis is characterized by hyperkeratosis-induced thickening of the skin, scaling of the skin, and inflammation [39]. The hereditary forms of ichthyosis are due to mutations on one or both alleles of over 30 identified genes that are primarily expressed in the upper epidermis [39]. Ichthyosis vulgaris, bullous ichthyosiform erythroderma, and ichthyosiform erythroderma are all hereditary forms of the disease [40]. As with epidermolysis bullosa, there is a noninherited form of ichthyosis, termed acquired ichthyosis (AI) [41]. AI is characterized by the scaling of skin and hyperkeratosis common to all forms of ichthyosis [41]. The exact causes of AI are unclear, as there are many contributing factors that may cause it. 

Atopic dermatitis (AD) is an autoimmune skin disease. The chronic inflammation characteristic of this skin disease is thought to be caused by defects in the innate immune system [42]. It is characterized by a disruption of the barrier function of the epidermis that results in dry skin that appears as eczematous patches and plaques [43].

Psoriasis is another example of an autoimmune skin disease. It is characterized by chronically red patches, irritated scaling, papules, and plaques caused by epidermal hyperplasia and altered differentiation of keratinocytes [44]. Psoriasis is thought to occur through a multifaceted interaction between genetic factors and environmental factors, making it neither truly acquired nor hereditary in origin [45]. 

Scleroderma is also categorized as an autoimmune skin disease characterized by the pathological remodeling of connective tissues [46]. Like psoriasis, scleroderma is influenced by both hereditary and acquired factors and does not fall clearly into either category [47].

Pemphigus is an uncommon autoimmune skin disease that causes blistering of the skin and buccal cavity [48,49]. Pemphigus is caused by autoantibodies targeted against antigens present on the exterior of the epidermal keratinocytes. This leads to disruption of cell–cell adhesion and results in formation of blisters. There are two primary forms of pemphigus, pemphigus vulgaris and pemphigus foliaceous. Pemphigus, like scleroderma and psoriasis, is an autoimmune disease that originates from a combination of hereditary and acquired factors [50].

Many forms of skin cancer, including melanoma, are acquired diseases. Melanoma is characterized by a malignant transformation of melanocytes to a state of uncontrolled proliferation [51]. Exposure to ultraviolet (UV) light has been shown to induce this transformation [52]. Several genetic risk factors have been identified for development of melanoma; however, it is not a hereditary disease [53].

The gene BCL11A, also known as (COUP-TF)-interacting protein 1 (CTIP1), is one genetic factor that is currently being investigated because of its role in the pathogenesis of epidermal barrier defects associated with several inflammatory skin disorders, namely AD, ichthyosis, and psoriasis [54,55,56,57]. BCL11A is expressed in the epidermis of mouse and human skin, as well as the hair follicles of developing and adult murine skin [16]. Analysis using immunohistochemistry (IHC) has revealed that BCL11A and hair follicle stem cell marker keratin 15 (K15) colocalizes in the hair follicle bulge region in murine skin (unpublished data). Germline deletion of BCL11A resulted in epidermal permeability barrier defects accompanied by significantly compromised skin terminal differentiation and altered skin lipid composition. This suggests that BCL11A regulates epidermal homeostasis in developing murine skin. Expression of CTIP1 homolog CTIP2 was dramatically increased in the skin epidermis of AD and allergic contact dermatitis (ACD) patients [58,59,60]. Mice with an epidermal specific deletion of CTIP2 (Ctip2^ep−/−^ mice) exhibited epidermal barrier defects and developed AD [17,58,59]. Furthermore, in vivo ablation of CTIP2 in epidermal keratinocytes significantly delayed skin wound healing and demonstrated its critical role in migration, proliferation, and differentiation of keratinocytes [60]. 

## 4. Current 3D Bioengineered Models of Skin Disease

Three-dimensional bioengineered skin models have been used to aid in the ongoing investigation of numerous skin diseases. This section details the various methodologies that have been used to model these diseases and improvements to be made on these models for future studies. The use of induced pluripotent stem cells (iPSCs) in 3D HSE skin disease models is also discussed. Initial investigation comparing 3D HSEs containing iPSC-derived keratinocytes and fibroblasts to 3D HSEs containing NHKs and NHFs has shown the potential for iPSC-derived keratinocytes and fibroblasts in future 3D HSEs [61]. Table 1 summarizes the studies discussed in this section.

### 4.1. Epidermolysis Bullosa Models

Several forms of EB, EBA, recessive dystrophic epidermolysis bullosa (RDEB), and Herlitz junctional epidermolysis bullosa (H-JEB) have been modeled using bioengineered 3D HSE models. In RDEB, the role of latent TGF-β signaling activation was studied using an HSE containing different combinations of normal healthy fibroblasts (NHFs), normal healthy keratinocytes (NHKs), patient-derived RDEB fibroblasts (RDEFs), and patient-derived RDEB keratinocytes (RDEBKs) [12]. This model was generated by manually depositing a combination of collagen I and fibroblasts (NHFs or RDEBFs) with keratinocytes (NHKs or RDEBKs) seeded on top to form a bilayer model. The layout for this model is shown in Figure 2 [12]. 

A similar 3D model was used to investigate the role of LAMB3 mutation on laminin 332 in H-JEB [14]. Laminin 332 is a structural protein responsible for skin adherence that is commonly affected by nonsense mutations in genes. These nonsense mutations cause abnormal, short, or diminished production of this protein in patients with H-JEB. A 3D HSE model was generated using a combination of NHFs, NHKs, H-JEB fibroblasts (H-JEBFs), and H-JEB keratinocytes (H-JEBKs). Like the model previously mentioned for investigation of RDEB, this model was generated by manual deposition of fibroblasts (NHFs or H-JEBFs) embedded in collagen type I followed by seeding of keratinocytes (NHKs or H-JEBKs) on top [76]. A thin layer of fibronectin was deposited at the junction between the dermis and epidermis prior to deposition of the epidermal layer to enhance cell attachment. This model was used to investigate the efficacy of gentamicin as a treatment for H-JEB. The results of this study showed gentamicin’s efficacy in treating H-JEB by restoring the H-JEB cells’ ability to produce functional laminin 332.

Induced pluripotent stem cells (iPSCs) have also been used in 3D EB HSEs. One iPSC-containing model used patient-specific iPSCs (PS-iPSCs) RDEB-derived keratinocytes to form the epidermal compartment of the construct [63]. This 3D HSE was constructed similarly to the prior RDEB 3D HSEs mentioned, using manual deposition of collagen type I and NHFs to form the dermal compartment. Either NHKs or RDEB PS-iPSC-derived keratinocytes were then seeded on top. Analysis of the constructs containing RDEB PS-iPSC-derived keratinocytes showed a lack of laminin 5 expression at the dermoepidermal junction (DEJ). The RDEB constructs also failed to express collagen type VII. These findings, along with appropriate expression of keratin 1 (K1) and locrin in non-RDEB iPSC-derived 3D HSEs, indicate the potential of this platform in future investigations of EB and treatment development. Another 3D HSE using iPSCs was used to investigate iPSC-derived fibroblasts as a potential treatment for RDEB [61].

These models of EB provided a relevant representation of the disease in a 3D skin-like environment; however, they did not illustrate the impact of EB on key skin adnexa and skin layers. Blisters characteristic of JEB form at the DEJ, which was only partially reconstructed in these models. Full reconstitution of the DEJ could be accomplished in future EB and JEB models and would improve the models’ efficacy [34]. Large melanocytic nevi can also form in patients with EB, which further complicates the treatment of this disease [77]. Inclusion of melanocytic nevi in future EB models would allow more accurate modeling and effective treatment development for patients with this aspect of EB. As was presented in the prior descriptions of skin diseases, there is also an acquired form of EB (EBA) that is an autoimmune form of the disease [78]. Prior investigation of EBA has not utilized an immune-competent model, which would allow for investigation into the role of the immune system in this disease and increase the relevancy of the modeling system used.

### 4.2. Ichthyosis Models

Ichthyosis has also been modeled using 3D skin equivalent platforms. The most common form of ichthyosis observed, ichthyosis vulgaris (IV), is characterized by filaggrin null mutations [79]. Several filaggrin knockdown 3D HSEs have been used to investigate the role of filaggrin in human skin and potential treatments for filaggrin null mutations, such as those found in IV. Filaggrin is believed to contribute to the integrity and mechanical properties of the stratum corneum. One 3D HSE used to investigate filaggrin knockdown was constructed using keratinocytes with filaggrin silenced using RNA interference technology by small interfering RNA (siRNA) [67,68]. This model was constructed by manually depositing a mixture of collagen type I and fibroblasts and then seeding NHKs with filaggrin knockdown using small interfering RNA (siRNA). This filaggrin-deficient model displayed a loss of keratohyalin granules and impairment of lamellar body formation. These findings were like those generated using human epidermal equivalents (HEEs) containing primary cells from IV patients [67,80,81]. These results were also consistent with prior results using the epidermis of IV patients [82]. Increased permeability of the epidermal barrier by a hydrophilic dye was also observed, further indicating the efficacy of this platform as an IV-like 3D HSE.

A more severe form of ichthyosis, harlequin ichthyosis (HI), has also been studied using 3D HSEs [69]. This model was designed to specifically investigate the role of the lipid transporter ATP-binding cassette A12 (ABCA12) gene, which causes HI. CRISPR/Cas9 was used to create an ABCA12 knockout keratinocyte cell line for this model. The dermal compartment of this 3D HSE was constructed using a mixture of collagen type I Matrigel laden with NHFs and THP-1 cells. The epidermal compartment, consisting of Crisper/Cas9 ABCA12 knockout N/TERT keratinocytes, was then seeded on top of the dermal compartment in a cloning ring. Analysis of this model revealed dysregulation of keratinocyte differentiation in the epidermis analogous to HI skin. Upregulation of proinflammatory cytokines comparable to that in HI skin was also observed. Treatment of this model with tofacitinib showed improvement in lipid barrier formation, further indicating the model’s relevance for HI investigation.

Current modeling of ichthyosis using 3D HSEs has focused on genetic knockout of genes involved in the pathogenesis of the disease using simplistic 3D HSEs. While the genes investigated thus far have provided valuable insight into ichthyosis, there are still several key genes of interest that have yet to be investigated using 3D HSEs. BCL11A would be a relevant addition for future investigations because of its role in epidermal homeostasis [16,18,19]. Further studies of ichthyosis using either iPSC-derived cells, PS-iPSC-derived cells, or primary ichthyosis patient-derived cells would also be beneficial for full comprehension of the disease and treatment development. Prior investigation of other skin diseases has benefited from the inclusion of these cells [61,82]. Investigation of acquired ichthyosis using an immunocompetent model would also be beneficial, as this form of ichthyosis involves autoimmunity.

### 4.3. Atopic Dermatitis Models

Three-dimensional HSEs have also proved useful in the study of AD. The company MatTek Inc. has generated a commercially available, full-thickness AD-like model called EpiDermFT [70]. To generate this model, MatTek treated a full-thickness model of human skin with a mixture of Th2 cytokines to induce a phenotype typical of AD. Histological and histochemical analysis of this model yielded findings consistent with AD. Corticosteroid treatment of this model also showed a partial ability to restore normal epidermal morphology and reversal of AD biomarker expression, which further illustrated the potential of this model as a tool for study of AD.

Addition of vasculature to HSEs has also been used in AD studies [71]. In one such study, a reconstructed epidermal compartment was generated using Transwells coated with collagen and NHKs deposited on top. The dermal equivalent of the HSE was bioprinted. A customized Transwell insert was constructed for the bioprinting of this component using an electrospun biodegradable scaffold. The scaffold consisted of poly(lactic-coglycolic acid) (PLGA) nanofibers. A customized biocompatible polycabrolactone O-ring was also glued to the bottom of the Transwell using a biocompatible adhesive to prevent any potential leakage of the bioprinted dermal compartment. These custom Transwell inserts were then treated with fibronectin the day before use and oxygen plasma 30 min prior to use. To print the actual vascularized dermal compartment, a mixture of fibrinogen, Novogel component 2, and aprotinin was used to form a hydrogel bioink. Before printing fibroblasts, induced pluripotent stem cells (iPSCs) and pericytes were suspended in this hydrogel. Fifteen minutes after printing, a dermal medium containing thrombin was added to allow partial fibrinogen polymerization. Following incubation, keratinocytes were seeded on top of the dermal compartment. To generate the AD 3D vascularized HSE, T_h_2 cytokine IL-4 was added to media during the air–liquid interface period of construct generation. To investigate the effect of anti-AD compounds in this model, three JAK inhibitors were used (tofacitinib, baricitinib, and ruxolitinib). Functional validation of the barrier function of these constructs was performed using high-throughput transepithelial resistance (TEER) measurements. This test illustrated the high barrier integrity of the vascularized 3D HSEs without IL-4 treatment. In the vascularized constructs treated to show AD-like pathogenesis, key AD phenotypes were observed, including spongiosis-like intercellular spaces, epithelial hyperplasia, and impaired differentiation [71]. Expression of keratinocyte pericellular E-cadherin and loricrin was also significantly reduced compared with control. Additionally, TEER measurement of AD-like vascularized 3D HSEs showed significant differences compared with control. The JAK inhibitors selected for this study were able to restore epidermal morphology and barrier function and increase expression of differentiated proteins and dermal–epidermal junction protein integrin β1. Taken together, this model was able to show significant efficacy for the study of AD.

An AD-like 3D HSE was also used to investigate the effect of dipotassium glycyrrhizinate (KG) on AD [15]. The HSE was constructed using a mixture of NHFs and collagen I with NHKs seeded on top. To create an AD-like HSE, HSEs were given media with IL-4 and IL-13 for 4 days. This treatment caused AD-like features to appear in the HSEs, including spongiosis-like intercellular spaces between cells, reduced filaggrin upon staining compared with control, and reduced expression of keratin 1 and keratin 10 proteins when compared with controls. mRNA and protein levels of AQP3 were also increased compared with controls, which is consistent with AD. To investigate the effect of KG on these AD-like HSEs, KG was administered along with IL-4 and IL-13 during the last 4 days of air–liquid interface culturing. Treatment with KG was able to block the formation of the spongiosis-like intercellular spaces seen in the AD-like HSEs and restore reduced expression of filaggrin. KG was also able to reduce expression of the AQP3 gene and levels of the proinflammatory cytokines IL-6 and IL-8. These findings indicate the potential of KG as a treatment for AD.

What these existing models of AD lack is the integration of patient-derived AD cells and relevant immune system components. Patient-derived AD cells would allow researchers to gain a better understanding of the in vivo response of AD-affected cells to various potential treatments. Addition of immune system components to future AD models would also increase the efficacy of future AD HSEs as accurate models for AD because of the inflammatory nature of the disease [43]. Furthermore, extended use of the standardized, commercially available model of AD available through MatTek would be advantageous to future studies of AD because of the potential of this model to produce more standardized results. Further investigation of both the genetic causes of AD and the overlap in genetic and phenotypic abnormalities presented when comparing AD and IV would also provide greater insights into both the pathways involved in AD and IV and potential treatments that could be developed [13,70,71,79]. As was suggested in the prior section on ichthyosis, future studies of AD would also benefit from investigation of the role of BCL11A in AD pathogenesis.

### 4.4. Psoriasis Models

Psoriasis has been widely studied using 3D HSEs. MatTek Life Sciences has a commercially available 3D model of psoriasis that has been used for a variety of studies on psoriasis [64,65,66]. The MatTek 3D psoriasis model is cultured in a custom cell culture insert using NHKs and psoriatic fibroblasts (PF) harvested from psoriatic lesions. The MatTek psoriasis model has been used to investigate topical RNA interference (RNAi)-based and topical small interference RNA (siRNA)-based treatments for psoriasis [63]. Both gene therapies showed potential to reduce the expression of several psoriasis-specific markers.

Another psoriasis-like model was generated through stimulation of a 3D HSE with interlukin-17A (IL-17A), interlukin-22 (IL-22), and tumor necrosis factor α (TNFα) [13]. The 3D skin model used in this study was manually generated by seeding a mixture of collagen I and NHFs on cell culture inserts, then seeding NHKs on top. This model was used to observe the ability of tofacitinib to prevent the formation of psoriasis-like morphology in the HSE through inhibition of JAK1/3. 

A more complex psoriatic 3D skin model was generated by incorporating T cells into a 3D HSE to create an immunocompetent skin model for psoriasis [66]. A schematic of this model’s construction is shown in Figure 3. This model used a psoriatic polarized human skin construct (pHSC) created through the incorporation of polarized Th1/Th17 cells or CCR6^+^ CLE^+^ T cells that were derived from cells from psoriasis patients. To create the immunocompetent psoriatic HSE, CD4^+^ T cells were integrated into the bottom of the HSEs. The HSEs themselves were created with a dermis composed of fibroblasts and collagen I, with keratinocytes seeded on top to form the epidermis. For the immunocompetent aspect of the model, CD4^+^ T cells were cultured overnight on collagen I gel, and then the fully differentiated HSEs were transferred onto this T cell–collagen gel. This construct allowed investigation of the immune interactions involved in psoriasis at both the disease level and the patient level. The inclusion of T cells in this psoriatic model decidedly improved the relevance of the modeling system, as psoriasis is thought to be caused by activation of the cellular immune system [83].

The commercial availability of a standardized psoriatic model is advantageous for the study of psoriasis as it ensures reproducibility of the model between different studies. The previously detailed examples of psoriatic skin models, however, lacked several key features that would enhance their efficacy as models of the disease. In addition to T cells, dendritic cells are thought to be involved in the pathogenesis of psoriasis. The vasculature of psoriatic patients is also markedly dilated, with the endothelial cells becoming activated in psoriatic lesions. Leukocytes may also gain entry to the skin via migration through these vessels. Inclusion of additional immune components, leukocytes and dendritic cells, and vasculature would enable researchers to better predict the in vivo response of psoriatic skin to treatment. Additionally, a BCL11A genetic knockout model would be beneficial to future studies of psoriasis due to the implication of BCL11A in epidermal barrier impairment [16]. There are also no current 3D HSEs using iPSC- or PS-iPSC-derived cells to investigate psoriasis; this tactic could prove useful to future research because of the versatility of this technology [84].

### 4.5. Scleroderma Models

Scleroderma has also been modeled using HSEs. A scleroderma model was generated using HSEs transplanted on skin severe combined immunodeficiency mice (skin-SCID) [72]. To generate the HSE, fibroblasts and keratinocytes were harvested from skin biopsies of both healthy donors and systemic sclerosis (SSc) donors. To generate a plasma scaffold for this model, plasma from healthy volunteers was combined with either healthy or sclerotic fibroblasts and allowed to clot into a 3D hydrogel. Keratinocytes from either healthy or SSc donors were then seeded on top. These skin constructs were then engrafted onto the backs of skin-SCID mice and allowed to integrate for up to 24 weeks. The scleroderma skin grafts transiently retained their phenotype in vivo, with features of scleroderma persisting up to 16 weeks following engraftment. This model was also used to confirm that the activation of fibrosis in vivo by autoantibodies occurs through platelet-derived growth factor receptor (PDGFR).

Another model of scleroderma was used to investigate the role of plasmacytoid dendritic cells (pDCs) in scleroderma [73]. To generate this HSE, NHFs and collagen I were combined, with NHKs seeded on top. This HSE was supplemented with supernatants from pDC cells previously treated with various forms of stimulation designed to either produce or not produce interferon (IFN). This model was used to show that BDAC2 targeting of pDCs could suppress the induced IFN response in human skin cells, suggesting that pDC inhibition could be a potential treatment for scleroderma. Scleroderma is characterized by an autoimmune response that generates fibrotic tissue, lending a definite relevance to the inclusion of dendritic cells in any scleroderma model [85]. 

The prior models for scleroderma lacked vasculature and various immune system components, such as T cells. Integration of vasculature in scleroderma HSEs would increase the platforms’ relevance, as the vasculature of scleroderma patients is also affected by the pathogenesis of this disease through the phenomenon of vascular endothelial cell injury.

### 4.6. Melanoma Models

Melanoma has been modeled using various 3D HSEs. One such HSE, containing melanoma spheroids, was developed for the study of melanoma in a relevant in vitro environment [75]. For this HSE, NHFs in collagen I were used with NHKs seeded on top. Melanoma spheroids were generated using a “hanging drop” method by pipetting droplets of a melanoma cell suspension onto a cell culture dish lid [74]. The melanoma spheroids were integrated into the HSE by including them in the dermal compartment of the construct. This model provided a reproducible method for screening of potential melanoma drugs.

Another melanoma HSE model was used to investigate melanoma cell migration, proliferation, and invasion using two different human melanoma cell lines, WM35 and SK-MEL-28 [75]. WM35 was used as a representative of the early phase of melanoma, which is typically confined to the epidermis. SK-MEL-28 was used as a representative of the later phase of melanoma, wherein the disease spreads to the dermis. The HSE used in this study was generated using deepidermized dermis (DED) prepared from human skin tissue. NHFs and NHKs were then seeded onto the papillary side of the DED to generate the healthy skin model. To construct melanoma HSEs, the same protocol was followed, with the two melanoma cell lines included in the initial seeding of NHKs and NHFs on the DED. This model enabled observation of the radial growth phase (RGP) of melanoma, using the WM35 cells, and the vertical growth phase (VGP) of melanoma, using the SK-MEL-28 cells. 

A significantly more complex melanoma 3D HSE was generated by integrating both vasculature and lymphatic capillaries into the HSE [86]. This model was formed by seeding fibroblasts in tissue culture plates with peripheral paper anchors and ascorbic acid to allow formation of manipulatable cell sheets. A schematic illustrating the process used to construct this model is shown in Figure 4. Human microvascular endothelial cells (HMVECs) were then seeded on two sheets of fibroblast cells. Melanoma spheroids were subsequently added on a third fibroblast cell sheet. Several different types of melanoma cells were used to form these spheroids so that different stages of melanoma could be observed. Lymphatic endothelial cells (LECs) and blood endothelial cells (BECs) were simultaneously deposited onto the sheet forming the surface of the construct. Keratinocytes were later added to the cell sheets containing the melanoma spheroids. These three cell sheets were then stacked and cultured. Using this model, two distinct capillary networks, vascular and lymphatic, could be observed. Chronic treatment with vemurafenib was used on WM983A and WM983B melanoma spheroid-containing HSEs. Both cell lines were expected to be sensitive to vemurafenib treatment, which was consistently reproduced using this model. Both the presence of blood and lymphatic vasculature and the relevant sensitivity to vemurafenib treatment indicate that this human melanoma model is an excellent candidate for further studies of melanoma. 

Addition of vasculature to models for melanoma has enabled researchers to better investigate the pathogenesis of these diseases. When melanoma metastasizes through the skin’s vasculature to distant sites, it becomes significantly more difficult to treat, and five-year survival rates drop to 23% for stage IV patients [87]. This makes vasculature in melanoma a feature of particular interest. The addition of lymphatic capillaries to melanoma models is also of significant interest, as this is another method through which cancers typically metastasize to distant sites. What these models lack is modeling of the interaction between melanomas and the immune system. The investigation of how some forms of melanoma are able to avoid detection by the immune system could improve future immunotherapy treatments for melanoma [88]. Hair follicles have also been implicated in the pathogenesis of metastatic melanoma [89,90,91]. Melanocyte stem cells (MeSCs) are present in the bulge region of the hair follicle and have been implicated as a potential cellular origin for melanoma [92]. Inclusion of hair follicles in melanoma 3D HSEs could provide additional insights into the complex pathogenesis of melanoma.

## 5. Methods for Generating 3D Human Skin Equivalents

There are several methods that have been developed for generation of 3D HSEs. The three discussed in this section are manual deposition, 3D bioprinting, and skin-on-a-chip. Manual deposition in this context refers to a researcher manually constructing a 3D HSE by depositing composites of biomaterials and cells into the desired vessels. While this is the simplest and most inexpensive method for construction of 3D HSEs, it has numerous drawbacks. Manual deposition is not readily reproducible; results may vary between different samples, researchers, and research groups. It is also difficult to precisely control the size and geometry of a construct using manual deposition, which hinders the researcher’s ability to successfully integrate adnexal structures into a 3D HSE constructed via manual deposition. Manual deposition also lacks high throughput capabilities, hindering its applications as a platform for drug development and testing [93,94].

An advantageous alternative to manual deposition in skin disease modeling is 3D bioprinting [95]. Three-dimensional bioprinting allows for a high degree of spatial and temporal control that allows researchers to generate complex constructs in a precise and reproducible manner. There are three primary categories of 3D bioprinters: inkjet, microextrusion, and laser-assisted bioprinters [96]. 

Inkjet bioprinters use either thermal, acoustic, or piezoelectric forces to eject drops of liquid onto a substrate. Thermal inkjet bioprinters work by electrically heating the print head such that enough pressure is generated to force droplets from the nozzle [96,97]. Acoustic inkjet bioprinters contain a piezoelectric crystal that generates acoustic waves inside the print head. These waves then disperse the bioink inside into droplets at set intervals. Inkjet bioprinting has the advantage of having very high throughput [98]. The disadvantage of inkjet bioprinting is the chance of reducing cell viability due to exposure to thermal and mechanical stresses. Inkjet bioprinters are also limited in that they cannot handle highly viscous liquids without greatly increasing the risk of cell death due to the increased force required to eject the bioink from the printer nozzle.

Microextrusion bioprinters work by providing automated and controlled extrusion of a material that is deposited onto a surface through a microextrusion head. Microextrusion forms continuous beads of material instead of the liquid droplets formed by inkjet bioprinters. A mechanical or pneumatic dispersion system is used for a microextrusion bioprinter. Specialized print heads with temperature-controlled cartridges are available for microextrusion bioprinters, enabling printing of temperature-sensitive biomaterials such as collagen [99]. Microextrusion bioprinters are also capable of handling more viscous liquids than inkjet bioprinting and laser-assisted bioprinting (LAB).

LAB applies the principles of laser-induced forward transfer. An LAB generally consists of a pulsed laser beam; a focusing system; a strip that has a donor transport support, typically made from glass coated with a layer used to absorb the laser energy; a layer of liquid biological material; and a receiving substrate that faces the ribbon. The LAB uses focused laser pulses directed at the absorbing layer to create a high-pressure bubble that then pushes the biologic materials toward the collector substrate [96]. The lack of a nozzle in LAB is advantageous, as it prevents the issue of nozzle clogging due to high cell density. LAB can also print at high cell density without affecting cell viability. However, LAB does have a lower flow rate than other 3D bioprinting technologies, as well as the potential to leave metallic residues on printed materials [98].

The emergence of lab-on-a-chip technology has allowed for the creation of skin-on-a-chip, another form of bioengineered skin. Skin-on-a-chip is a highly versatile platform for the study of skin and investigation of skin diseases [100]. Skin-on-a-chip models typically involve the use of a small compartment containing some mixture of fibroblasts, keratinocytes, and other cells to form adnexa. One study utilizing skin-on-a-chip technology created a vascularized, perfusable model of human skin [101]. A schematic illustrating this skin-on-a-chip is shown in Figure 5. Another skin-on-a-chip model was developed as platform for drug development, specifically as it pertained to inflammation and edema in skin. Addition of hair follicles and complex adnexal structures have also been accomplished through skin-on-a-chip models [102]. Skin-on-a-chip technology provides significant advantages in the field of skin bioengineering in terms of the level of complexity possible and high throughput capabilities. One of the primary disadvantages of skin-on-a-chip is its inability to be used for grafting purposes. While it is an excellent platform investigation of drugs for skin disease treatment, it is not a suitable platform for the development of skin graft alternatives. Some diseases affecting skin, such as diabetes, require skin grafts as a form of disease treatment [103].

## 6. Challenges in Engineering 3D Skin Models

To provide an ideal platform for investigating skin diseases, a reproducible and complex 3D HSE that accurately recapitulates the complex biomolecular and biomechanical microenvironments of skin in both a healthy state and a diseased state is required. This endeavor presents significant engineering challenges. While there has been considerable success in generating more simplistic, manually generated 3D HSEs, these models typically lack key features of skin relevant to the diseases they are modeling. To generate more physiologically relevant models of skin diseases, a reproducible method for generating these models that integrates adnexa is necessary. As was discussed previously, 3D bioprinting is an excellent alternative to manual deposition that provides greater reproducibility [95]. Commercially available 3D bioprinters allow researchers to have greater precision and spatial control when generating 3D HSEs. Three-dimensional bioprinters also allow for greater control when adding adnexal features such as hair follicles and vasculature [104,105,106]. HSEs containing hair follicle-like structures have been generated using microextrusion-based bioprinting. Microextrusion-based bioprinters have also been used to generate printed channels lined with ECs or HUVECs for formation of microvasculature in 3D HSEs. LAB has been used to create a pattern of ECs onto a collagen biopaper containing embedded MSCs. Advancements in skin-on-a-chip technology have also provided a means for addressing challenges presented in skin bioengineering. These advancements include the integration of perfusable vasculature in a skin-on-a-chip model [101].

Relevant adnexa for specific skin disease representation are also lacking in most models. To model a wide variety of skin diseases, a reproducible method for integrating hair follicles, circulatory vasculature, lymphatic vasculature, sebaceous glands, and sweat glands must be developed. Each skin adnexum presents a unique challenge to integrate into a 3D HSE, as each adnexum has its own unique cell population and structure. The vasculature, which provides nutrients for the skin, has been studied extensively for a variety of purposes, including application in skin grafts for burn victims and those with diabetic ulcers [107]. The size of the microvasculature found within the dermis presents a challenge to recreation. Arterial capillaries found in the microvascular network of the skin have external diameters of 10 to 12 μm and internal diameters of 4 to 6 μm [26]. Extrusion-based bioprinting has a much lower printing resolution limit than this, at approximately 200 μm [108]. Inkjet-based bioprinting has a higher resolution; however, at 30 μm, it is still too low to accurately recreate microvasculature [108]. Laser-based bioprinting also does not have a small enough printing resolution at 50 μm [108]. Cutaneous lymphatic vessels present a similar problem to skin microvasculature. While cutaneous lymphatic vessels can reach up to 60 μm in size, this is still quite small. Inkjet and laser-based bioprinters have potentially high enough resolution to print these lymphatic vessels; however, neither bioprinting system has the capability of handling the high-viscosity fluids, such as collagen type I, that are typically used when printing the dermal compartment of an HSE.

Sweat glands and sebaceous glands present both similar and different challenges to those posed by the cutaneous circulatory and lymphatic systems. The eccrine sweat gland is composed of a coiled tubular structure that reaches from the dermis to an epidermal opening on the surface of the skin [109]. Each individual secretory tubule of the eccrine gland ranges in size from 30 to 40 μm [110]. The apocrine sweat gland has a similar structure to that of the eccrine sweat gland; however, it has a larger tubule size ranging from 80 to 100 μm [110]. Recreating these structures in vitro presents the challenge of creating a 3D structure that is both coiled and tubular while also generating a fine enough tubule. Sebaceous glands have a layered structure, similar to that of the epidermis, containing specialized keratinocytes called sebocytes that go through a terminal differentiation process extending to the outer layer of the gland [111]. The challenge in recreating this adnexum in an HSE is the induction of this layered differentiation.

Hair follicles are arguably the most complex of the skin adnexa, as they contain a particularly diverse, layered population of cells and are a reservoir for stem cells [27]. Hair itself is composed of trichocytes, or terminally differentiated keratinocytes. The shaft from which the hair originates is termed a hair follicle, and together with the sebaceous gland, apocrine glands, and arrector pili muscle, it forms the pilosebaceous unit. Development of the hair follicles involves coordinated ectodermal–mesodermal interactions. The hair follicle stem cells of ectodermal origin create the epithelial components of the hair follicle, including the sebaceous gland and apocrine gland. The mesodermal-derived cells form the follicular dermal papilla and a connective tissue sheet. Neural crest-derived melanocyte progenitors create the hair follicle pigmentary unit. To accurately recreate this adnexal structure, all these units must be recreated using relevant cell types. Hair cycling is also a feature of interest in several skin diseases, including psoriasis, ichthyosis, and atopic dermatitis [18]. To investigate hair cycling in vitro using a 3D HSE, constructs must be cultured for sufficient periods to allow for hair cycling to occur. This is currently limited by the period in which 3D HSEs can support cell viability in vitro. Addition of complex, functional vasculature to a 3D HSE could potentially address this problem by increasing the lifetime of 3D HSE in vitro.

## 7. Current Advances in the Field of Skin Engineering

The ability of researchers to create complex modeling systems has increased significantly, from simplistic 2D models to complex 3D models. Researchers are now able to generate skin models containing multiple cell types, a stratified epidermis, a dermal compartment, and various skin adnexa. Integration of immune system and nervous system components has also been accomplished in several models. Among the skin adnexa that have been recreated are hair follicles, sweat glands, sebaceous glands, circulatory vessels, and lymphatic vessels. Table 2 presents a summary of different methods used for generating these 3D HSEs, cells used in them, and any adnexa included.

To create immunocompetent skin models macrophages, dendritic cells (DCs), Langerhans cells (LCs), and CD4^+^ T cells have been included in various platforms [112,113,114]. One immunocompetent HSE was generated by creating a dermis composed of collagen, Jurkat T cells, and fibroblasts with an epidermis composed of keratinocytes. Another immunocompetent HSE was generated by including pDCs into an immunocompetent model of scleroderma, as described in the prior section on current diseased skin models [73].

While there has not been as much progress made in integrating a bioengineered nervous system into an HSE, there has been progress made in generating neural equivalents in other platforms. A 3D-bioprinted, electrically conductive bioink with human neural stem cells (hNSCs) was used to create a nervous system model [115,116]. The scaffold used for this model was composed of agarose, chitosan, and alginate. This construct was electrically stimulated using a two-electrode setup designed to stimulate the hNSCs. This neural cell laden biogel was able to form dense arrays of polarized neuronal cells exhibiting axons and dendritic arborizations.

Another study concerning nervous system recapitulation used a variety of bioinks to determine the best combination for a construct that contained encapsulated Schwann cells with dorsal root ganglia (DRG) neurons seeded on top [117]. The authors found that a combination of either alginate, HA, and fibrinogen (40FAH) or RGD modified alginate, HA, and fibrinogen (FRAH) best accomplished this. They observed that the Schwann cells within the hydrogels were able to guide the growing direction of the neurite outgrowth from the DRG neurons cultured on the scaffold surface. 

A more advanced model featuring both immune system and nervous system components was created in a full-thickness skin model [118]. A hypodermis consisting of a lipoaspirate scaffold and coated on both sides with hiNSCs in a collagen gel scaffold was used in this model. A schematic illustrating the construction of this model is shown in Figure 6. The lipoaspirate scaffold consisted of adipocytes, preadipocytes, endothelial cells, and smooth muscle cells donated from abdominoplasty. This HSE was formed using a silk–collagen hydrogel and fibroblasts with keratinocytes seeded on top. qRT-PCR was performed using the macrophage markers CD68 and CSF1 to confirm the presence of immune cells in the HSE. Proinflammatory markers IL-6 and RANTES were also used to analyze the presence of immune cells. Both macrophage markers and proinflammatory markers increased from week 1 to week 6, while the adipose marker ACRP30 decreased. This analysis indicated successful differentiation of adipose cells into macrophages over the construct maturation timeline. To evaluate the development of nerves within the HSE over time, the hiNSCs were predyed before inclusion in the construct. Analysis of the dyed cells indicated that hiNSCs were able to remain in the hypodermis for up to 6 weeks and form dense neural networks.

Hair follicles present an interesting challenge for researchers. A sophisticated construct for hair follicle growth was generated using a collagen–glycosaminoglycan (C–GAG) matrix, skin-derived precursors (SKPs), and epidermal stem cells (Epi-SCs) [119]. SKPs were seeded on C–GAG matrices and cultured alone in precursor experiments, which determined that SKPs expressed hair follicle-inductive genes in the C–GAG matrix. SKPs and Epi-SCs were seeded into the C–GAG matrixes and implanted onto full-thickness excisional wounds of BALB/c nu/nu mice. Three weeks after implantation, the wound sites demonstrated de novo hair genesis originating from the implanted Epi-SCs and SKPs. 

Another hair follicle generation model used human adult scalp dermal progenitor cells and epidermal stem cells to create a bilayered HSE that exhibited hair follicle formation [120]. To form this HSE, an initial acellular layer of collagen was generated, and a second collagen matrix containing dermal progenitor cells was then deposited on top of it. These HSEs were then grafted onto the full-thickness wounds of nude mice. The HSEs were implanted at different stages of in vitro development (5, 9, 14, and 21 days of culture) to determine whether different stages of growth influenced results. The early-stage HSEs showed hair follicle formation with sebaceous glands.

A more complex HSE was created containing both hair follicles and vasculature using 3D-printed molds to allow the physiological arrangement of cells within the hair follicle [106]. A schematic illustrating this mold and the HSE constructed using it is shown in Figure 7. The dermal compartment of this construct was generated by seeding type I collagen mixed with fibroblasts and GFP-tagged HUVECs onto a 3D-printed mold designed to mimic the spatial relationship of hair follicles. Following polymerization, the 3D-printed molds were removed, and the construct was cultured to allow for capillary formation. Dermal papilla cells (DPCs) were then seeded on top. The following day, keratinocytes were seeded to form the epidermal compartment. Culture of HSEs for 3 weeks resulted in elongated hair follicles with observable organization of inner and outer root sheaths, as confirmed through analysis using K5 and AE13 markers. To study the in vivo formation of hair follicles and vasculature using these constructs, athymic mice were used. A full-thickness wound was made on the back of each mouse, and a hat-shaped silicon chamber was placed underneath the mouse skin. The HSEs generated previously were placed through the hole at the top of the mold and maintained in culture for 5 days. Following this, the chamber was removed, and the HSEs were sutured onto the mouse. Induction of human hair growth was observed after 4 to 6 weeks. 

Sweat glands have also been integrated into HSEs with some success. In one model, a combination of bioprinting and spheroid cultures was used to create HSEs with sweat glands and hair follicles [121]. To create the sweat glands, an alginate–gelatin gel was combined with a single-cell suspension of MSCs and a previously made tissue suspension obtained from the plantar skin dermis. This mixture was then bioprinted and cultured in sweat gland induction culture media. To introduce hair follicles to this construct, hair follicles were first formed in spheroid cultures. Keratinocytes and fibroblasts were mixed in hair follicle induction culture medium and pipetted as droplets onto the top of a cell culture dish lid. Hair follicle spheroids were then collected and seeded into the sweat gland-containing scaffolds previously constructed. After full development of the sweat gland- and hair follicle-containing scaffolds, several markers were investigated to confirm the presence of an inner and outer layer of sweat glands as well as sweat ducts and hair progenitor markers for hair follicles. Analysis of these markers indicated the presence of differentiated sweat glands and hair follicle progenitors.

An HSE containing only sweat glands was generated using a 3D extracellular matrix (ECM) composed of a hydrogel laden with dermal homogenates from mouse papillary dermis (PD) [122]. The bioprinted construct was a mixture of the PD homogenates and mouse epithelial progenitor cells. To determine if the epithelial progenitor cells successfully differentiated after culture, hereditary analysis of the samples was completed by harvesting total RNA from the HSEs and analyzing using real-time PCR. This analysis led to the conclusion that that the epithelial progenitor cells differentiated into sweat glands in vitro. To test in vivo formation of sweat glands, the construct was transplanted into the paws of wild-type C57/B16 mice at induced burn sites. To evaluate sweat gland-specific function, an iodine/starch sweat test was completed 14 days later. This test revealed that sweat glands had formed in vivo.

Some progress has also been made in generating engineered sebaceous glands. An HSE containing human induced pluripotent stem cells (hiPSCs) was used to generate sebaceous glands [123]. To generate this model, hiPSCs and fibroblasts were integrated into a porous scaffold constructed of chitosan cross-linked C–GAG polymer with keratinocytes seeded on top. Analysis of this construct revealed that the hiPSC-derived sebocytes self-organized into 3D organoids that expressed key sebaceous gland droplets, as well as displaying accumulated lipid droplets.

Reproducing a vascular network within an HSE has been a subject of keen interest, as having a nutrient supply to the construct allows for better recapitulation of in vivo diseases and extends graft life posttransplant in vivo. A vascularized and perfusable HSE was generated using human keratinocytes, fibroblasts, pericytes, and endothelial cells [124]. The dermal compartment of the construct consisted of type I collagen, fibroblasts, endothelial cells, and pericytes. The dermal compartment was first cultured to allow formation of endothelial networks to occur. The epidermal compartment consisted of keratinocytes in media printed on top of the dermal compartment. Immunohistochemical analysis of the HSEs revealed the presence of human CD31^+^ vessel-like structures located within the dermis. Pericytes were also found to increase the thickness and maturity of the epidermis and induce expression of laminin 5 at the epidermal–dermal junction. These HSEs were also tested in vivo through implantation on the backs of athymic mice following 8 days of in vitro culture. Injection of fluorescent UEA-1 confirmed perfusion of the vasculature formed by the implanted HSEs.

An HSE composed of keratinocytes, fibroblasts, and human microvascular endothelial cells (HMVECs) in a gelatin alginate composite hydrogel was used to generate another, vascularized HSE [125]. This HSE was composed of a mixture of fibroblasts, HMVECs, and hydrogel with keratinocytes seeded on top. To assess the in vivo performance of this skin graft, nude mice were used. Full-thickness wounds were made on the backs of the mice, and HSEs were grafted onto the wound sites. Four weeks postsurgery, the grafts were analyzed, and significant angiogenesis was observed. It was also confirmed that the microvessels formed in the grafts were derived from the HMVECs that has been printed.

Another approach to integrating vasculature into HSEs used a skin-derived decellularized ECM (S-dECM)-based bioink [126]. Porcine skin was obtained, washed extensively to remove all cells and debris, lyophilized. The actual HSE used in this study to create a vascularized construct was a skin patch containing the S-dECM bioink, adipose-derived stem cells, and endothelial progenitor cells. This construct was cultured for 3 days prior to implantation to allow for prevascularization of the skin patch. To assess the in vivo functionality of the HSE, BALB/cA nu/nu mice were used. Full-thickness wounds were made on the backs of the mice, and HSEs were grafted onto the wound sites. To determine if the skin patch had successfully generated microvasculature, in vivo blood flow measurement and histological and immunohistology analysis of the samples were performed. The skin patch showed formation of CD31^+^ vessels and significant neovascularization. The in vivo blood flow analysis showed that the skin patch did increase blood flow over time.

A skin-on-a-chip model also succeeded in generating perfusable vasculature, as was discussed previously in Section 5. This model used keratinocytes, fibroblasts, and HUVECs in a skin-on-chip model with microfluidic channels [101].

Addition of lymphatic vasculature to 3D HSEs has also received attention, often in combination with circulatory vasculature. A model containing features of both the circulatory and lymphatic systems was generated using lymphatic endothelial cells (LECs) [127]. To create the ECM used for the dermal compartment of this construct, fibroblasts were grown for 21 days, allowing them to form an ECM sheet. LECs were included in the model by seeding them on the fibroblast sheet and allowing them to grow and generate a microvasculature network. To form the overall construct, two sheets of fibroblasts and LECs were layered under a sheet with fibroblasts alone. Immunostaining of the resultant construct with Prox-1, a marker for LEC differentiation, showed differentiation of LECs. The resultant vessels also proved to be CD31^+^.

Another HSE with both blood and lymphatic capillary networks was generated using a multilayered cell accumulation technique that involved coating the cells with fibronectin–gelatin (FN–G) nanofilms. This technique was used with fibroblasts to generate a layered dermis [127]. To add blood and lymphatic-like capillary networks, HUVECs and normal human dermal lymphatic microvascular endothelial cells (NHDLMECs) were coated with FN–G nanofilms and seeded onto the surface of the layered fibroblasts. Additional fibroblasts coated with FN–G nanofilm were then seeded onto the construct. Following 7 days of incubation, dense blood and lymph capillary networks were observed. The epidermis was formed by first coating the dermal layer with type IV collagen and then seeding keratinocytes on top. The resultant HSEs were immunoassayed for CD31 and LYVE-1, which are markers for endothelial cells and lymphatic cells, respectively. This staining revealed two distinct capillary networks, a blood capillary network, and a lymphatic capillary network.
pharmaceutics-14-00319-t002_Table 2Table 2Current 3D HSE models with adnexal structures and methods used for their construction.MethodAdnexal Structure(s)Cells UsedReference(s)3D Bioprinting, ExtrusionVasculatureIL-4-Treated NHKs, iPSCs, NHFs, pericytes[71]Manual DepositionImmune SystemMUTZ-LC, NHFs, NHKs[128,129,130]Manual DepositionImmune SystemNHFs, NHKs, LCs, DCs[131]Manual DepositionImmune SystemNHFs, NHKs, MUTZ-3-LCs[132]Manual DepositionImmune SystemNHFs, NHKs, DCs[133]Manual DepositionImmune SystemNHFs, NHKs, Macrophages[134]Manual DepositionImmune SystemNHKs, NHFs, Peripheral Blood Mononuclear Cells, CD4^+^ T cells[135]Skin-On-A-ChipVasculatureHaCaT Cells, HS27 Fibroblasts, HUVECs[94]3D Bioprinting, ExtrusionNervous SystemhNSCs[115]Manual DepositionNervous SystemhNSCs[116]3D Bioprinting, ExtrusionNervous SystemSchwann Cells[117]Manual DepositionImmune System, Nervous SystemNHKs, NHFs, hiNSCs[118]Manual DepositionHair FollicleSKPs, Epi-SCs[119]3D Bioprinting, ExtrusionHair Follicle, Sweat GlandNHKs, NHFs, MSCs[120]Manual DepositionHair FollicleDermal Progenitor Cells, Epi-SCs[121]Manual DepositionVasculature, Hair FollicleDPCs, NHKs, NHFs, HUVECs[107]3D Bioprinting, ExtrusionVasculatureNHKs, NHFs, Pericytes, Endothelial Cells[124]3D Bioprinting, ExtrusionSweat GlandEpithelial Progenitor Cells[122]Manual DepositionSebaceous GlandhiPSCs[123]3D Bioprinting, ExtrusionVasculatureNHKs, NHFs, HMVECs[125]3D Bioprinting, ExtrusionVasculatureAdipose-Derived Stem Cells, Endothelial Progenitor Cells[126]Skin-On-A-ChipVasculatureNHKs, NHFs, HUVECs[101]Manual DepositionLymphatic System, VasculatureNHFs, HUVECs, NHKs, NHDLMECs[127]Manual DepositionLymphatic System, VasculatureLECs, NHFs[127]

## 8. Future Directions

Extraordinary progress has been made in the field of bioengineered efficacy models of skin disease in the last decade. Techniques used for generating 3D HSEs have become increasingly sophisticated. More researchers are generating 3D HSEs using 3D bioprinting and skin-on-a-chip technology as opposed to simple manual deposition. This shift towards more sophisticated technology will allow researchers to better generate highly precise and sophisticated models for future investigation of skin diseases. Complex 3D HSEs containing one or more adnexal structures are increasing in prevalence, showing a clear advancement in the field towards more biomimetic, efficacious bioengineered skin models. There have also been marked improvements in the generation of these adnexal structures, with several key adnexa such as hair follicles and vasculature receiving considerable attention. Use of 3D HSEs to study skin disease has also become more common.

Despite the remarkable advances that have been made in the field of skin disease modeling, there are still several key areas that have thus far been underserved. While a variety of models have been made that recapitulate different disease states, from psoriasis to melanoma, study of the genetic aspects of wound healing remains a key area for future studies. Specifically, the genetic aspects of wound healing play a key role in several skin diseases, including atopic dermatitis and ichthyosis vulgaris, and contribute to their pathogenesis. One gene of interest for future research that has been discussed throughout this paper is BCL11A, which is thought to play a role in epidermal barrier integrity. Lack of epidermal barrier integrity is a key feature in many skin diseases, including ichthyosis, psoriasis, and atopic dermatitis. Future studies of skin diseases should focus more on genetic factors affecting a wide variety of skin diseases, such as BCL11A.

There are also several skin diseases that have yet to be investigated using 3D HSE models. Pemphigus, discussed prior in Section 3, is one such skin disease. Some skin diseases that have been modeled using 3D HSEs also have disease subtypes that have not been investigated using 3D HSEs yet. Acquired ichthyosis is one such disease. Future investigations of skin disease using 3D HSEs should place more emphasis on investigating diseases that have not been modeled using 3D HSEs yet, as use of these modeling systems may reveal further insights into the disease than previously utilized 2D in vitro models.

Evolving technology has enabled researchers to create increasingly complex models of skin for the investigation of skin disease. While it is still common practice to generate more simplistic models containing a dermal–epidermal bilayer using manual deposition methods, the field is heading towards more complex models that take advantage of 3D bioprinting technology. The commercial availability of 3D bioprinters with highly tunable printing parameters has allowed researchers to generate increasingly sophisticated models with greater reproducibility. The addition of adnexa to skin disease models is a crucial next step in skin disease modeling and will continue to advance the ability of the field to identify potential therapeutics. The inclusion of vasculature in more HSEs will allow for greater efficacy in skin disease modeling through its ability to increase culture time and its relevance in many skin diseases. Looking forward, researchers should strive to include multiple adnexa relevant to the skin disease being modeled, as well as any relevant immune and neural system components. More advanced skin models will also be applicable to understanding the mechanisms involved in skin homeostasis and wound healing, enabling researchers to develop more targeted therapeutics to assist in wound healing.

## Figures and Tables

**Figure 1 pharmaceutics-14-00319-f001:**
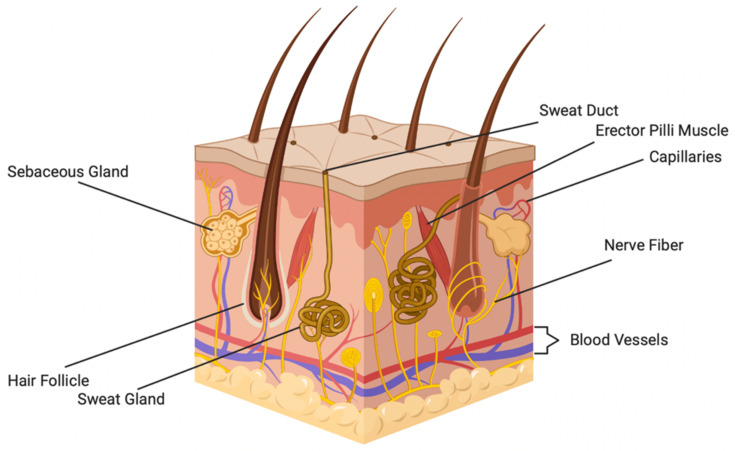
Prominent skin adnexal structures. Adapted from “Anatomy of the Skin” by Biorender.com (accessed on 15 November 2021). https://app.biorender.com/biorender-templates.

**Figure 2 pharmaceutics-14-00319-f002:**
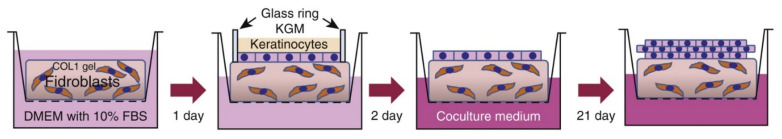
Schematic illustrating construction of an RDEB 3D HSE used for investigation of the role of latent TGF-β signaling activation in RDEB. Adapted with permission from [12], published by Nature Publishing Group, 2021.

**Figure 3 pharmaceutics-14-00319-f003:**
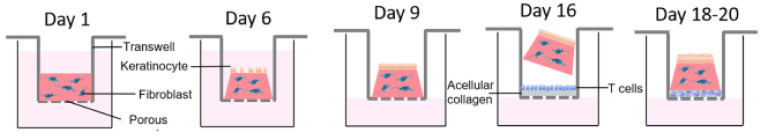
Schematic showing construction of a psoriasis model incorporating T cell infiltration. Adapted from [66], published by Nature Portfolio, 2020.

**Figure 4 pharmaceutics-14-00319-f004:**
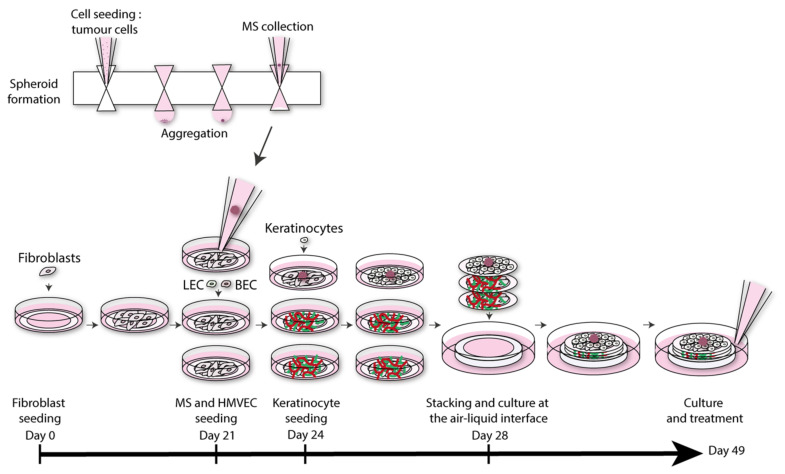
Schematic showing construction of a melanoma model with vasculature and lymphatic vessels. Adapted from [86], published by Nature Portfolio, 2018.

**Figure 5 pharmaceutics-14-00319-f005:**
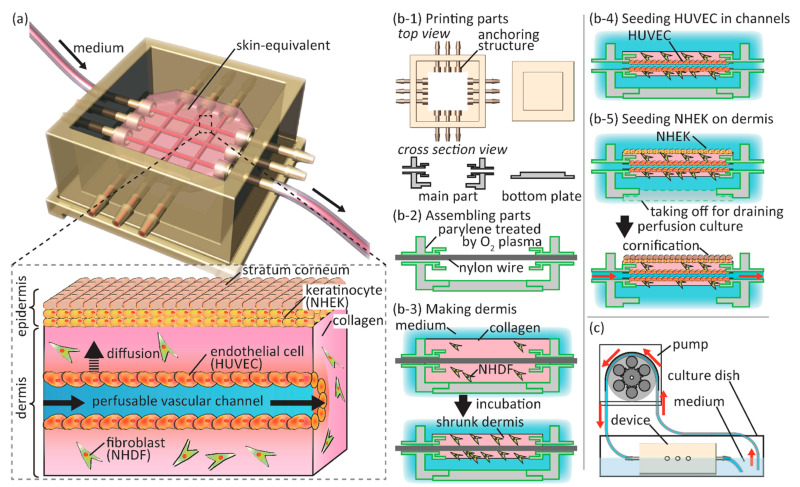
Schematic showing a vascularized, perfusable skin-on-a-chip model. (**a**) Illustration depicting the HSE and culture chamber. (**b1**–**3**) Fabrication of the culture chamber and HSE. (**c**) Perfusion system consisting of the culture chamber, peristaltic pump, and silicone tubes. Adapted with permission from [101], published by Elsevier, 2017.

**Figure 6 pharmaceutics-14-00319-f006:**
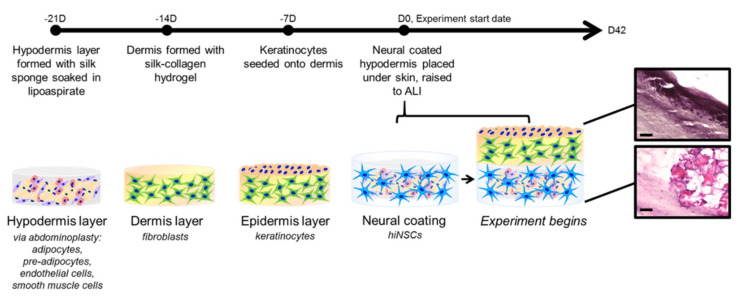
Construction of an HSE with a nervous system and hypodermis. Adapted with permission from [118], published by Elsevier, 2019.

**Figure 7 pharmaceutics-14-00319-f007:**
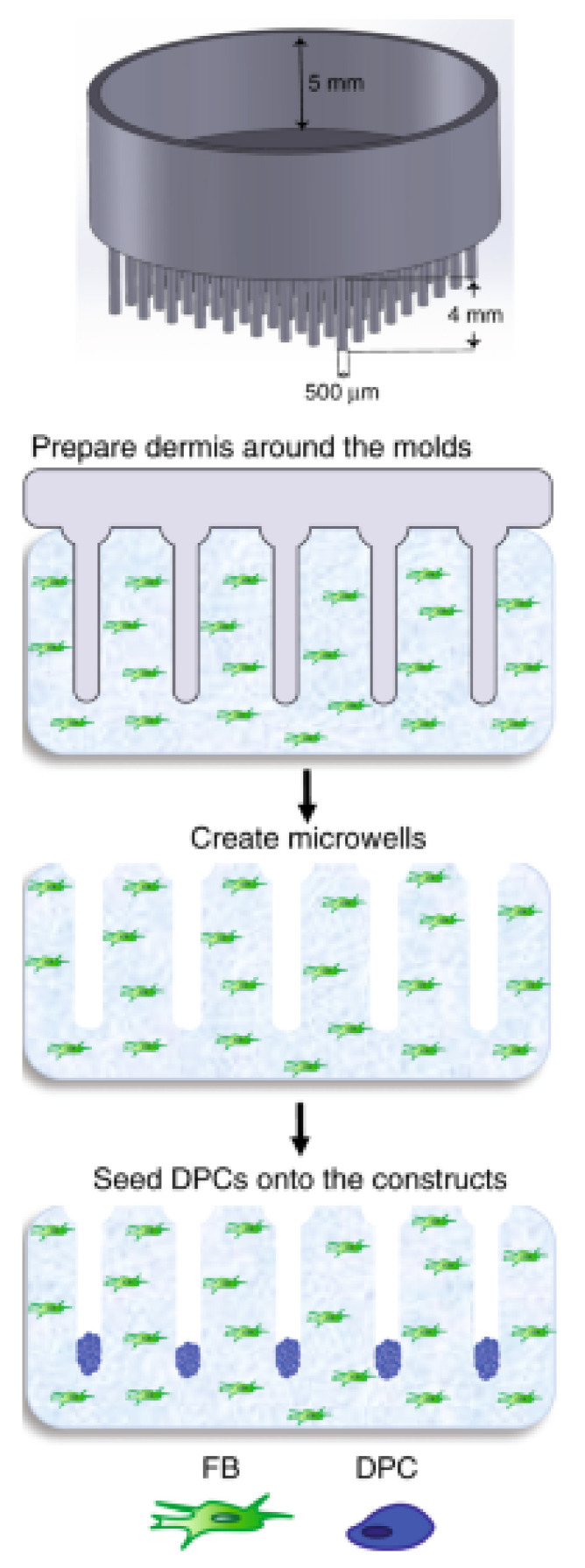
Schematic showing 3D-printed mold used for construction of a vascularized, hair follicle-containing 3D HSE. Adapted from [106], published by Nature Portfolio, 2018.

**Table 1 pharmaceutics-14-00319-t001:** Skin diseases that currently have 3D HSE models.

Skin Disease	Method	Cells Used in Model	Reference
Recessive Dystrophic Epidermolysis Bullosa	Manual Deposition	RDEBKs, RDEBFs, NHKs, RDEB PS-iPSC-Derived Keratinocytes, RDEB PS-iPSC-Derived Fibroblasts	[12,62]
Herlitz Junctional Epidermolysis Bullosa	Manual Deposition	H-JEBKs, H-JEBFs, NHKs, NHFs	[14]
Psoriasis	n/a	Psoriatic Fibroblasts, NHKs	[63,64,65]
Psoriasis	Manual Deposition	IL-17A-, IL-22-, and TNFα-Treated NHKs and NHFs	[13]
Psoriasis	Manual Deposition	Polarized Th1/Th17 cells, CD4^+^ T cells, NHKs, NHFs	[66]
Ichthyosis Vulgaris	Manual Deposition	siRNA Filaggrin Knockdown Keratinocytes, NHFs	[67,68]
Harlequin Ichthyosis	Manual Deposition	CRISPR/Cas9 Knockdown ABCA12 N/TERT Keratinocytes, NHFs, THP-1	[69]
Atopic Dermatitis	n/a	Th2 Cytokine-Treated NHKs, Th2 Cytokine-Treated NHFs	[70]
Atopic Dermatitis	3D Bioprinting,	IL-4-Treated NHKs, iPSCs, NHFs, Pericytes	[71]
Atopic Dermatitis	Manual Deposition	IL-4- and IL-3-Treated NHFs, IL-4- and IL-3-Treated NHKs	[15]
Scleroderma	Manual Deposition	Patient-Derived SSc Fibroblasts, Patient-Derived SSc Keratinocytes, NHKs, NHFs	[72]
Scleroderma	Manual Deposition	pDCs, NHFs, NHKs	[73]
Melanoma	Manual Deposition	NHFs, NHKs, 451-LU	[74]
Melanoma	Manual Deposition	WM35, SK-MEL-28, NHFs, NHKs	[75]

## Data Availability

Not applicable.

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
