# Peer review of "Bioengineered Efficacy Models of Skin Disease: Advances in the Last 10 Years"

_pharmaceutics, 2022, doi:10.3390/pharmaceutics14020319_

Round 1
Reviewer 1 Report
The comment for Authors:
It is interesting that the authors want to describe the bioengineered efficacy skin models in the last ten years. However, the quality of this paper is not suitable for the publish of Pharmaceutics now. It should be minor revised by the following comments.
- The article is well organized and written fluidly.
- In sections 4-6, it strongly suggests that the authors should cite the figures of current models from the references. It will help the reader more clearly understand the structure of models.
- In section 6, the authors only mentioned the 3D bioprinting technology for fabricating the skin models. However, in addition to the 3D bioprinting, there are many skin models presented in other articles such as Biomaterials 2017 Mori et al.; Scientific Reports 2016 Lee et al.; Lab on a chip 2013 Lindner et al. The authors should describe more the current bioengineered models and describe the advantages and challenges of these models.
Reviewer 2 Report
This manuscript described rather deep with excellent examples the principles and considerations of 3D models of skin, the basic knowledge of skin cell biology is combined with bioengineering and regenerative medicine approaches. But, if my goal of this review is mainly the attempt to improve the manuscript, I dare to make several critical comments.
There are several errors or printing errors:
Line 69 the comma is needed between “touch” and “temperature”
Line 317 Hitological
Other several type-mistakes and absent commas were in several places.
The description of the diseases chosen is large enough and written well. Although I suggest to add the description of Pemphigus as well. If there are no skin equivalents for it, so probably it is worthwhile to mention this fact. Ichthyosis is mentioned in the review, but the authors weirdly did not cite the papers, in which such skin equivalents were described. For example, https://www.ncbi.nlm.nih.gov/pmc/articles/PMC6562513/
https://www.ncbi.nlm.nih.gov/pmc/articles/PMC7456239/.
The description of skin equivalents from induced pluripotent stem cells seems non-sufficient (maybe should be really represented widely in the review), although they were created for the diseases, for EB as well.
https://www.ncbi.nlm.nih.gov/pmc/articles/PMC3102348/
https://www.ncbi.nlm.nih.gov/pmc/articles/PMC3795682/
It seems to me, that the “table” would be the best representation for the skin diseases and the respective models known at present.
As the review makes the accent on the 3D bioprinting, it would be desirable to create the table with already printed equivalents together with indications of printing technology, cell types used and the structures, which were reproduced.
Reviewer 3 Report
The paper provides a review of models for skin disease. The topic is appropriate for the journal and useful, however as written the article is not suitable for publication.
The paper does a good job summarizing previous and current research in the field, but is lacking in providing insights that aggregate the works in a new way, which is evident by the lack of figures and tables in the manuscript. Additionally, there is inconsistency in how topics are reviewed, there are long sections for instance for Section 4 where as the bioprinting section 6 is very short for being an important part of the paper.
Specific comments are as follows:
-Many references could include a greater number of authors included in the references section. There are too many references with just the first author followed by et al that it makes it difficult to determine contributions from authors throughout the references.
-The abstract contains references which is not typical for this journal.
-The two figures included in the paper Figure 1 and 2 are basic anatomy, it would be useful to see figures representing the different models or bioprinting methods. Additionally, Tables that aggregate data across references in a meaningful way could also improve the manuscript.
-Page 6 and Page 7 have paragraphs that are over about a page or more long which need to be more focused.
-Section 5 begins with high-level motivation that may be more suitable for the introduction, additionally the introduction has a low number of references and motivation for setting up the review and rationale for the organization of the review.
-Section 6 is very brief and underdeveloped for bioprinting being a key aspect of this paper. For instance the LAB is briefly summarized in how it works but the importance of LAB and its contributions are not described well.
-The future descriptions section at the end of the manuscript seems to build further on current advances rather than provide a full overview of where the field is and specular where it could go in the future, this is evident by much of the text here being summaries of new citations that weren't referenced earlier in the manuscript.
Reviewer 4 Report
This manuscript provides an overview of recent progress in the biofabrication of 3D human skin equivalents, with a special focus on the development of model tissues that mimic hereditary or acquired skin diseases. This article is interesting, up to date, and very well written.
I only have two minor suggestions for potential improvements:
1. A few illustrations could have been included in Sections 4 and 8 to make them more attractive. Of course, this would require asking permission to reprint certain figures or tables from the cited references. Most publishers provide a straightforward procedure for this via the "Rights & Permissions" link, often handled by RightsLink.
2. The bibliography might be extended with the following references:
Mathes SH, Ruffner H, Graf-Hausner U. The use of skin models in drug development. Advanced Drug Delivery Reviews. 2014;69–70:81-102.
Varkey M, Visscher DO, van Zuijlen PPM, Atala A, Yoo JJ. Skin bioprinting: the future of burn wound reconstruction? Burns & Trauma. 2019;7.
Kim BS, Ahn M, Cho W-W, Gao G, Jang J, Cho D-W. Engineering of diseased human skin equivalent using 3D cell printing for representing pathophysiological hallmarks of type 2 diabetes in vitro. Biomaterials. 2021;272:120776.
Round 2
Reviewer 1 Report
The authors replied all my questions.
Reviewer 3 Report
The authors have vastly improved the manuscript and it is acceptable for publication after proofing, the references still appear in the abstract and it could still be good to include one of the two basic anatomy figures early in the paper for unfamiliar readers.
